# Bacterial α-Glucan and Branching Sucrases from GH70 Family: Discovery, Structure–Function Relationship Studies and Engineering

**DOI:** 10.3390/microorganisms9081607

**Published:** 2021-07-28

**Authors:** Manon Molina, Gianluca Cioci, Claire Moulis, Etienne Séverac, Magali Remaud-Siméon

**Affiliations:** Toulouse Biotechnology Institute (TBI), Université de Toulouse, CNRS, INRAE, INSA, 135, Avenue de Rangueil, CEDEX 04, F-31077 Toulouse, France; manon.m.molina@gmail.com (M.M.); gianluca.cioci@insa-toulouse.fr (G.C.); moulis@insa-toulouse.fr (C.M.); e_severa@insa-toulouse.fr (E.S.)

**Keywords:** glucansucrase, branching sucrase, GH70, lactic acid bacteria, sucrose-active enzymes

## Abstract

Glucansucrases and branching sucrases are classified in the family 70 of glycoside hydrolases. They are produced by lactic acid bacteria occupying very diverse ecological niches (soil, buccal cavity, sourdough, intestine, dairy products, etc.). Usually secreted by their producer organisms, they are involved in the synthesis of α-glucans from sucrose substrate. They contribute to cell protection while promoting adhesion and colonization of different biotopes. Dextran, an α-1,6 linked linear α-glucan, was the first microbial polysaccharide commercialized for medical applications. Advances in the discovery and characterization of these enzymes have remarkably enriched the available diversity with new catalysts. Research into their molecular mechanisms has highlighted important features governing their peculiarities thus opening up many opportunities for engineering these catalysts to provide new routes for the transformation of sucrose into value-added molecules. This article reviews these different aspects with the ambition to show how they constitute the basis for promising future developments.

## 1. Introduction

Polymers are used as basic materials for the manufacture of innumerable daily life products, as well as more sophisticated material in medicine, diagnostics and fine chemistry. Most of them are derived from petroleum-based chemistry. Environmental and sustainability concerns push forward the research on renewable and recyclable bio-derived structures showing equivalent, new or improved physico-chemical properties. Among bio-sourced polymers, polysaccharides from plant, algae, fungi or microbial organisms are considered nowadays as promising options to replace petroleum-based products with new architectures. Many of them, especially plant and algal polysaccharides, are already important feedstocks for well-established bio-based industries [1]. The market for microbial polymers is more modest but continuously growing. Their production is independent of environmental and climatic fluctuations and downstream processing is generally easy, which is a significant advantage. However, despite a great diversity offering a strong potential for innovation, only a few of them—namely xanthan, gellan, hyaluronic acid, welan, clavan, fucopol, pullulan and dextran—are currently marketed [2,3,4]. Given the knowledge on polysaccharide synthetic pathways as well as the huge amount of available enzyme sequences and the progress made in protein and metabolic engineering, the supply of enzymatically or microbiologically produced polysaccharides, shaped on demand, is expected to rise significantly [5,6].

In this field, α-glucans produced by lactic acid bacteria (LAB) from different genera (mainly *Streptococcus, Leuconostoc, Lactobacillus, Weissela*) when grown on sucrose are particularly appealing targets [7,8,9]. These polymers participate in biofilm formation and can protect the bacteria against environmental stresses (e.g., dessication, biocides, antibiotics, phagocyte attack). They can also mediate the adhesion to surfaces [10]. In particular, α-glucans formed by the *Streptococcus* genus were well-studied due to their contribution to the dental plaque formation, the tooth surface colonization and the development of dental caries [11]. The panel of polymer structures characterized to date and obtained from this cheap and abundant agro-resource is broad. Dextran is mainly composed of α-1,6 osidic linkages in the main chain, but can also contain α-1,2/α-1,3 or α-1,4 in main chain or at branching points, mutan contains a majority of α-1,3 osidic bonds, reuteran mixes α-1,4 and α-1,6 osidic linkages and alternan is quasi-strictly composed of alternating α-1,3 and α-1,6 osidic bonds (Figure 1). The sizes, types and arrangements of α-osidic linkages in the polymers as well as the degree of branching are highly dependent on the producing bacterial strains and their enzymatic arsenal. They are the primary determinants of the physicochemical, biological, and mechanical properties of individual α-glucan, and govern their range of potential applications. The dextran produced by *Leuconostoc mesenteroides* NRRL B512-F (an α-1,6 linked α-glucan with very few α-1,3 branching) was the first microbial biopolymer commercialized since 1948 to be used as plasma substitute. For this medical application, only low molar mass fractions obtained after partial hydrolysis and solvent fractionation of the native high molar mass polymer are used [7,8,12,13,14]. Nowadays, the market still mainly concerns low molar mass α-1,6 linked linear dextrans for biomedical or analytical applications [15]. Other usages in food, cosmetics, nanotechnologies and/or mineral extractions have also been reported [14,16]. Moreover, the insolubility and mechanical properties of α-1,3 linked glucans that originated from *Streptococcus* species (produced with recombinant enzymes) have also recently raised attention to develop versatile α-glucan-based biomaterial [17,18].

The bacterial enzymes involved in α-glucan production from sucrose are called glucansucrases (GSs) and they can be assisted by branching sucrases (BRSs). Indeed, BRSs are specific for branching linear α-1,6-linked dextran to produce highly branched polymers (Figure 1) [19,20,21], and in the absence of dextran acceptor, BRSs almost exclusively catalyse hydrolysis. Both GSs and BRSs are classified in the family 70 of glycoside hydrolases (GH) [22]. They are naturally very efficient transglucosylases and do not require expensive nucleotide-activated sugars (NDP-sugar) as substrate for the synthesis of very high molar mass polymers (from 10^3^ to 10^9^ g/mol) [7,20,23,24,25,26]. In addition, they are rather promiscuous and catalyze transglucosylation from sucrose onto a broad range of acceptors (sugars or unnatural hydroxylated molecules), the recognition and efficiency of the reaction being dependent on the enzyme selectivity [26,27]. These properties are exploited to produce functional oligosaccharides [28] in particular isomaltooligosaccharides [29], oligoalternans (oligosaccharides with alternating α-1,6 and α-1,3 linkages) [30,31,32,33], oligoreuterans with α-1,6 and α-1,4 linkages [34] or α-1,2 branched gluco oligosaccharides [35,36,37,38,39] commercialized under the brand name BioEcolans^®^ (Solabia) for food, healthcare or cosmetic applications. Furthermore, GSs and BRSs can also be used to produce glucoderivatives such as flavonoid glucosides [40,41,42,43] or glyco-co-polymers [44,45]. In family GH70, other types of polymerases have been recently discovered i.e., the 4,6 and 4,3-α-glucanotransferases [46,47]. They also synthesize α-glucans but use α-1,4 linked glucans as glucosyl donor instead of sucrose. These enzymes will not be included herein, for more information the reader can refer to the excellent review of Meng et al. describing their potentialities and their evolutionary relationships with family GH70 sucrases and family GH13 amylases [48,49].

The amazing versatility of GSs and BRSs holds multiple assets for the development of new biosourced molecules targeting an even wider range of innovative applications. Nowadays, this potential is clearly amplified by the discovery of an increasing number of new enzymes, by the progress made in the understanding of their molecular mechanism and by engineering approaches that can be deployed to expand their synthetic capabilities. This review places the focus on these various aspects and intends to cover more specifically the most recent and exciting developments.

## 2. Inventory of Characterized GSs and BRSs

### 2.1. Native GSs and BRSs Produced by LAB

In 1861, Louis Pasteur was the first to demonstrate the microbial origin of the gelatinous substance observed in cane and beet-sugar juices, just before Scheibler discovered, a few years later, that the slimy substance was a polysaccharide, that was named “dextran” due to its positive rotary power [50,51]. In 1878, Van Tieghem et al. isolated the first dextran producing microorganism belonging to the species *Leuconostoc mesenteroides*, and Hestrin and co-workers named “dextransucrase” the extracellular enzyme responsible for dextran production from sucrose [52,53]. Dextran was the first α-glucan that found industrial applications as a blood plasma substitute in the 1940s, which motivated the search for other producing strains. In 1954, Jeanes and collaborators screened 96 strains of LAB and highlighted the impressive structural diversity of these homopolysaccharides, exclusively composed of glucosyl units and containing all possible α-1,6, α-1,4, α-1,3 and/or Q1,2 glucosidic bonds in main chains and/or at branching points [24,54]. 

Historically, GSs have been predominantly isolated from strains belonging to *Leuconostoc* (*Ln*), *Streptococcus* (*S*) and *Lactobacillus* (*Lb*, recently reclassified as *Latilactobacillus*, *Lentilactobacillus, Limosilactobacillus* sp. [55] and the characterization of GSs from these species is still very active. For example, GSs from *Lb. satsumensis* strains present in water kefir grains have been shown to produce a water-soluble dextran as well as an insoluble one rich in α-1,3 osidic linkages [56]. More recently, several dextransucrases produced by water kefir borne *Lb. hordei* TMW 1.1822 and *Lb. nagelii* TMW 1.1827 [57], and by *Lb. kunkeei* H3 and H25 isolated from honeybees were also characterized [58]. Furthermore, *Ln. lactis* SBC001 was used to produce gluco oligosaccharides showing prebiotic and anti-inflammatory properties by fermentation on sucrose and maltose-rich medium [34]. Moreover, GSs produced by species from different genera including *Pediococcus*, *Weisella* and *Oenococcus* sp. have also been isolated [59,60,61,62,63]. 

However, it is worth mentioning that LAB strains often produce several GSs showing different product specificities as confirmed by enzyme biochemical characterization [64,65,66], gene cloning [67] and genome sequencing [68,69,70,71,72]. For example, the inventory of the GH70 enzymes encoded by *Ln. citreum* NRRL B-1299 allowed the identification of 5 GSs and 1 BRS, namely DSR-A, DSR-B, DSR-E, DSR-DP, DSR-M and BRS-A [71]. Moreover, all of them are large enzymes (from 140 to around 300 kDa) often slightly degraded and associated to the cell membranes. This renders extremely challenging the individual characterization of each enzyme secreted in a crude fermentation medium. This is why since the 90s, recombinant production of GSs is preferred for structure–function studies. This trend is amplified by the fast development of genome sequencing over the last decade and the ever-growing number of putative GSs and BRSs in the databases.

### 2.2. Recombinant GSs and BRSs Produced by E. coli

In April 2021, 790 sequences were recorded in the GH70 family of CAZy database (compared to only 264 in November 2015 [48], see http:/www.cazy.org/GH70, accessed on 24 July 2021). However, less than 60 sequences of biochemically characterized enzymes are listed (Table 1). This number is very small, and many new and original enzymes are likely to be discovered in the future. To date, the database records 40 enzymes classified as dextransucrases, α-1,6 glucosyltransferases or α-1,6/α-1,3-glucosyltransferases (representing 77% of listed GH70 sucrose active enzymes); seven as mutansucrases or α-1,3 glucosyltransferases (13.5%); two as reuteransucrases (3.8%); two as alternansucrases (3.8%) and only one as a branching sucrase (1.9%) (Figure 1). Dextransucrases are still overrepresented in the family and, among them, the last ones characterized present very interesting properties. For instance, the recombinant enzyme DSR-M from *Ln. citreum* NRRL B-1299 produces a linear and low molar mass dextran directly from sucrose [23,71]. Additionally studied recently, recombinant dextransucrases from *Lb. reuteri* TMW 1.106 and *Ln. citreum* TMW2.1194 are distinctive in their ability to synthesize highly branched dextrans with O3 and O4-linked side chains [73,74]. Wangpaiboon et al. also identified a very original recombinant DEX-N glucansucrase of *Ln. citreum* ABK-1 that is responsible for the synthesis of two types of glucans from sucrose. The first one is a soluble dextran composed of 90% α-1,6 linkages while the second one is insoluble and harbors a block-wise pattern of α-1,3 and α-1,6 osidic bonds [75]. 

Branching sucrases were recently discovered by genome analysis of bacterial species known to produce highly branched dextrans e.g., *Ln. citreum* NRRL B-1299 and NRRL B-742. New genes showing sequence singularities compared to those coding for dextransucrases, alternansucrases and mutansucrases were revealed to encode BRSs specialized in dextran decoration by addition of α-1,2 or α-1,3 osidic branches [21,71]. Since then, data mining based on the search for specific motifs allowed the identification of other BRSs in the genomes of *Ln. fallax* KCTC3537 and *Lb. kunkeei* EFB6 [21]. These enzymes were proposed to constitute a subfamily in the GH70 family. It was recently suggested that the GS from *Ln. mesenteroides* BD3749 (Gsy) producing a glucan rich in α-1,6/α-1,4 and α-1,3 linkages and showing high sequence similarity with BRSs could be an evolutionary intermediate between BRSs and GSs [76]. 

Finally, two enzymes containing two catalytic domains both featuring the characteristics found in GH70 family have been described to date. In these enzymes, one domain shows GS activity and the other one BRS activity. This is the case of the DSR-E enzyme from *Ln. citreum* NRRL B-1299 in which the catalytic domains 1 and 2 (CD1 and CD2) code for a dextransucrase activity and an α-1,2 branching activity, respectively [77]. Similarly, the recently discovered gtfZ glucansucrase from *Lb. kunkeei* DSM 12361 also possesses a dextransucrase domain and an additional domain dedicated to dextran branching via α-1,3 osidic linkages [78]. 

**Table 1 microorganisms-09-01607-t001:** List of GH70 characterized sucrases recorded in the CAZy database in April 2021 (in bold) as well as few others reported in the literature. Enzymes are classified according to their specificity: R for reuteransucrases, M for mutansucrases, D for dextransucrases, A for Alternansucrases, BRS for branching sucrases. Enzyme size is indicated in number of amino acids (aa). When possible, other sucrases described in literature with their Genbank accession number—even if not listed in the GH70 family yet—are also indicated. NP = not published. * refers to enzymes with a crystal structure available.

Enzyme	Organism	Genbank	Specificity	Size (aa)	Reference
**GTF-0**	*Lb. reuteri*	AAY86923.1	**R**	1781	[79]
**GTF-A***	*Lb. reuteri*	AAU08015.1	**R**	1781	[80]
**ASR***	*Ln. mesenteroides*	CAB65910.2	**A**	2057	[81]
**ASR**	*Ln. citreum*	AIM52834.1	**A**	2057	(Wangpaiboon et al. NP)
**GTF-SI***	*S. mutans*	BAA26114.1	**M**	1455	[82]
**GFT-ML1**	*Lb. reuteri*	AAU08004.1	**M**	1772	[83]
**GFT-L**	*S. salivarius*	AAC41412.1	**M**	1449	[84]
**GTF-J**	*S. salivarius*	AAA26896.1	**M**	1517	[84]
**GTF-I**	*S. sobrinus*	BAA02976.1	**M**	1590	[85]
**GTF-I**	*S. downei*	AAC63063.1	**M**	1597	[86]
**GTF-B**	*S. mutans*	AAA88588.1	**M**	1475	[87]
GTF-I	*S. criceti*	BAF62338.1	**M**	1461	[88]
GTF-F	*S. orisuis*	BAF62337.1	**M**	1466	[88]
GFT-D	*S. mutans*	AAN58619.1	**GS (n.d)**	1462	[89]
GTF-C	*S. mutans*	AAN58706.1	**GS (n.d)**	1455	[89]
GTF-B	*S. mutans*	AAN58705.1	**GS (n.d)**	1476	[89]
**LcDS**	*Ln. citreum*	BAF96719.1	**D**	1477	[90]
**GTF-U**	*S. sobrinus*	BAC07265.1	**D**	1554	[91]
**GTF-S**	*S. downei*	AAA26898.1	**D**	1365	[92]
**GTF-R**	*S. oralis*	BAA95201.1	**D**	1575	[93]
**GTF-M**	*S. salivarius*	AAC41413.1	**D**	1577	[84]
**GTF-Kg3**	*Lb. fermentum*	AAU08008.1	**D**	1595	[94]
**GTF-Kg15**	*Lb. sakei*	AAU08011.1	**D**	1561	[94]
**GTF-K**	*S. salivarius*	CAA77898.1	**D**	1599	[84]
**GTF-I**	*S. sobrinus*	BAA14241.1	**D**	1575	[95]
**GTF-G**	*S. gordonii*	AAC43483.1	**D**	1577	[96]
**GFT-D**	*S. mutans*	AAA26895.1	**D**	1430	[97]
**GTF-33**	*Lb. parabuchneri*	AAU08006.1	**D**	1463	[94]
**GTF-1971**	*Lb. animalis*	CCK33644.1	**D**	1585	(Ruhmkorf et al., NP)
**GTF-180***	*Lb. reuteri*	AAU08001.1	**D**	1772	[94]
**GTF-1624**	*Lb. curvatus*	CCK33643.1	**D**	1697	(Ruhmkorf et al., NP)
**GTF-106A**	*Lb. reuteri*	ABP88726.1	**D**	1782	(Kaditzky et al., NP)
**DSR-X**	*Ln. mesenteroides*	AAQ98615.2	**D**	1485	[98]
**DSR-WC**	*W. cibaria*	ACK38203.1	**D**	1472	[99]
**DSR-T**	*Ln. mesenteroides*	BAA90527.1	**D**	1015	[100]
**DSR-S**	*Ln. mesenteroides*	AAD10952.1	**D**	1527	[101]
**DSR-P**	*Ln. mesenteroides*	AAS79426.1	**D**	1454	[102]
**DSR-N**	*Ln. mesenteroides*	AFP53921.1	**D**	1527	(Siddiqui et al. NP)
**DSR-K39**	*W. cibaria*	ADB43097.3	**D**	1445	[60]
**DSR-F**	*Ln. citreum*	ACY92456.1	**D**	1527	[103]
**DSR-D**	*Ln. mesenteroides*	AAG61158.1	**D**	1527	[104]
**DSR-C39-2**	*W. confusa*	CCF30682.1	**D**	1412	[59]
**DSR-C**	*Ln. citreum*	CAB76565.1	**D**	1477	[81]
**DSR-BCB4**	*Ln. citreum*	ABF85832.1	**D**	1465	[105]
**DSR-B742**	*Ln. citreum*	AAG38021.1	**D**	1508	[106]
**DSR-B**	*Ln. citreum*	AAB95453.1	**D**	1508	[107]
**DSR-A**	*Ln. citreum*	AAB40875.1	**D**	1290	[108]
**DEX-YG**	*Ln. mesenteroides*	ABC75033.1	**D**	1527	[109]
**DEX-T**	*Ln. citreum*	ACA83218.1	**D**	1495	[69]
**Wc392-DSR**	*W. confusa*	AHU88292.1	**D**	1423	(Krajala et al., NP)
**DSR**	*Ln. lactis*	ACT20911.1	**D**	1500	(Kim et al., NP)
**DSR-DP**	*Ln. citreum*	CDX66641.1	**D**	1278	[71]
**DSR-M***	*Ln. citreum*	CDX66895.1	**D**	2065	[71]
**wcCab3-DSR**	*W. confusa*	AKE50934.1	**D**	1401	(Shukla et al., NP)
DSR-R	*Ln. mesenteroides*	AAN38835.1	**D**	1330	(Kim et al. NP)
GTF-P	*S. sanguinis*	BAF43788.1	**D**	1575	[95]
GTF-Tl	*S. sobrinus*	AAX76986.1	**D**	1506	[110]
GTF-106B (DSR106.1)	*Lb. reuteri*	ABP88725	**D**	1883	[111]
DSR	*Lb. animalis*	CCK33644.1	**D**	1585	[111]
**DSR-E**	*Ln. citreum*	CAD22883.1	**D+α-1,2BRS**	2835	[77]
GtfZ	*Lb. kunkeei*	KRK22577.1	**D+α-1,3BRS**	2621	[78]
Gsy	*Ln. mesenteroides*	ANJ45894.1	**GS (n.d)**	1466	[76]
**BRS-A**	*Ln. citreum*	CDX66896.1	**α-1,2BRS**	1877	[71]
BRS-B	*Ln. citreum*	CDX65123.1	**α-1,3BRS**	1888	[21]
BRS-C	*Ln. fallax*	WP_010006776.1	**α-1,3BRS**	1774	[21]
BRS-D	*Lb. kunkeei*	WP_051592287.1	**α-1,2BRS**	1463	[21]
GBD-CD2*	*Ln. citreum*	CAD22883.1	**α-1,2BRS**	1694	[112]

## 3. Structure–Function Relationships

### 3.1. Catalytic Mechanism and Products

GSs and BRSs adopt a two-step α-retaining mechanism and catalyze the formation of a covalent β-glucosyl-enzyme intermediate from sucrose [113,114,115]. First, sucrose is accommodated in the active site with the glucosyl and fructosyl moieties in subsites −1 and +1, respectively, and according to the subsite numbering established by Davies et al. [116]. Then, an aspartate (the nucleophile) attacks the anomeric carbon of the glucosyl unit through the assistance of a glutamate (the acid-base catalyst) that gives its proton to the fructosyl moiety, leading to fructose release and formation of a covalent β-glucosyl-enzyme via an oxocarbenium transition state stabilized by a third highly conserved aspartate (the Transition State Stabilizer, TSS). The second step corresponds to the de-glucosylation where the hydroxyl group of an acceptor, accommodated in subsite +1 and activated by the deprotonated glutamate, intercepts the β-glucosyl-enzyme intermediate to be glucosylated. Different acceptor molecules can compete and the glucosyl transfer occurs either on (i) water molecules (hydrolysis reaction), (ii) the growing glucan chain, (iii) fructose resulting in sucrose isomers (leucrose: α-D-Glc*p*-(1→5)-β-D-Fru*p* and isomaltulose: α-D-Glc*p*-(1→6)-β-D-Fru*f*), (iv) sucrose itself or (v) an hydroxylated acceptor introduced in the reaction mixture (Figure 2) [117]. Reactivity of GSs and BRSs towards the sucrose donor substrate is quite stringent. These enzymes can use glucoside substituted with a good leaving group at the anomeric position (e.g., glucoside-fluoride or paranitrophenyl glucoside) but sucrose is by far their preferred glucosyl donor [118].

GSs are usually very efficient transglucosylases and hydrolysis is a minor reaction. In contrast, in the absence of exogenous dextran acceptor, hydrolysis is preponderantly catalyzed by BRSs. Transfer onto fructose is frequent, especially at the end of the reaction when fructose is in a large excess [119]. Glucosyl transfer onto sucrose was revealed for different GSs including DSR-S, ASR [119], DSR-M [23], GTF-S3 [120], and GTFA [121]. Isomelezitose (α-D-Glc*p*-(1→6)-β-D-Fru*f*-2 ↔1-α-D-Glc*p*), which could result from sucrose glucosylation, can also be produced [122]. Glucose, sucrose and its isomers can serve as initiator for the polymerization reaction yielding first oligosaccharides and then α-glucans [23,119]. Polymer growth was proposed to follow a semi-processive mechanism [119]. Both DSR-S from *Ln. mesenteroides* NRRL B-512 and ASR from *Ln. citreum* NRRL B-1355 were shown to initiate the polymer formation. Both DSR-S from *Ln. mesenteroides* NRRL B-512 and ASR from *Ln. citreum* NRRL B-1355 were shown to initiate the polymer formation by successive glucosyl transfer onto glucose or sucrose itself to produce oligosaccharides. At the beginning of the reaction, oligosaccharides are produced via a distributive mode of elongation (multi-chain). After a while, the elongation mechanism is shifted toward a more processive mechanism (single chain elongation) leading to very high molar mass polymer. The domain V of GSs is proposed to play an important role in the control of the shift between distributive and processive mechanisms by providing an anchoring zone for long polymer chain bindings (see section below). Different populations of low molar or high molar mass polymers may be obtained. The product profile is dependent on the transglucosylation versus hydrolysis ratio, and on the processivity of each enzyme. In some cases, the specificity and structure of the oligosaccharides produced in the first step may also prevent further elongation [119,123,124]. Notably, dextransucrase DSR-M and mutansucrase GTF-I produce only low molar mass polymers [23,125].

GSs can recognize a large panel of acceptors, whether disaccharides [27,126,127,128,129,130], trisaccharides [126], polyphenols [41,42,43,131,132,133], terpenoids [134,135] or alkyl-glucosides [136]. The natural promiscuity of GSs is variable depending on the enzyme. It has been exploited since the 60s to modulate the physico-chemical properties (solubility, HLB, redox potential), bioactivity (prebiotic, organoleptic, antioxidant properties) or bioavalability of acceptor molecules. Branching sucrases have not been tested yet with a broad range of acceptors. The α-1,2 and α-1,3 BRSs both recognize fructose as acceptor but not maltose that is one of the best acceptors for many GSs. They prefer α-1,6 linked glucosyl chains comprising at least four glucosyl units to introduce branch linkages and this is accompanied by a remarkable 80-fold increase of the reaction initial rate [21,78,112]. Recently, they were also shown to glucosylate flavonoids [40] and chemically protected disaccharides or tetrasaccharides offering new access to various oligosaccharides that could enter in the preparation of glycosidic vaccines against shigellosis [137,138,139]. 

### 3.2. Mechanistic Insights from Primary Structures

The first GS sequences from *S. mutans* GS-5 (GTF-B) [87], *S. mutans* LM7 (GTF-C) [140] and *S. sobrinus* MFe28 (GTF-I) [86] were released in 1987. Sequence alignments revealed the presence of a putative (β/α)_8_ barrel circularly permutated compared to those of the GH13 family enzymes [141]. Seven conserved regions, retrieved in both GH13 and GH70 families but ordered differently, were identified [142,143,144,145]. This enabled to locate the putative nucleophile (aspartate Asp635, ASR numbering), the acid/base catalyst (glutamate Glu673, ASR numbering) and the TSS (aspartate Asp767, ASR numbering) as well as highly conserved residues defining the subsite −1 of the GH13 enzymes (Figure 3). The GH70 enzymes were therefore placed in the same GH-H clan as the GH13 enzymes and were later joined by the GH77 family enzymes. The role of these key residues was further confirmed by mutagenesis studies. All the mutants targeting the amino acids of the catalytic triad almost totally lost activity. Mutation of the strictly conserved arginine of motif II in GtfR from *S. oralis* ATCC10557 (Arg633 of ASR) and the histidine of motif IV in GTF-I, DSR-S and GtfR (His766 of ASR) resulted in enzymes with 0.5% residual activity compared to the parental enzyme [100,146,147]. Mutations of the strictly conserved glutamine of motif I in GTF-180 and DexYG were less detrimental but increased hydrolysis reaction and led to mutants with around 3% and 34% of residual activity, respectively [148,149]. Finally, the strictly conserved aspartate of motif I (D1169 in ASR) was replaced by an alanine in GTFR-100 from *S. oralis*. This also drastically reduced activity (6% compared to the wild-type enzyme) and favored hydrolysis reaction [150].

The amino acids close to the nucleophile, acid/base catalyst or TSS also rapidly attracted attention due to their rather good conservation in enzyme sharing similar linkage specificity. For instance, an asparagine of motif II (Asn639 in ASR) is conserved in GSs sequences but not in BRSs. This residue is important for activity and hydrolysis in GTF180 but not critical in DexYG [149,152]. Mutation of the equivalent amino acid in the branching sucrase GBD-CD2 (obtained by truncation of DSR-E [77]) was detrimental for branching activity [153]. Similarly, the tryptophan from motif III (Trp675 in ASR) conserved in GSs is essential for transglucosylation. Indeed, its mutation to glycine or alanine almost totally inactivates GTF-I from *S. mutans* GS-5, GTF180 and DSR-M [147,154,155]. DexYG is the sole exception as its transferase and hydrolytic activities were not altered [149]. In ASR from *Ln. citreum* NRRL B-1355, ASR from *Ln. citreum* ABK-1 and GTF180, its replacement with another aromatic residue resulted in a decrease of activity but the transglucosylation activity was partially maintained [124,154,156]. To note, Gsy is the only GS lacking this conserved tryptophan [76]. The introduction of a tryptophan at a nearby position in GBD-CD2 mutants A2249W and G2250W did not promote transglucosylation [153].

### 3.3. Going Further with the Three-Dimensional Structures

The 3D-structure of GTF180-ΔN confirmed the hypothesis of the circular permutation [157,158] and revealed an organization in five distinct domains: the domains A, B and C, structurally close to those of the GH13 family enzymes, and the domains IV and V, unique to GH70 enzymes. Each domain is built of a non-contiguous chain except for domain C, which forms the base of the U-shape fold [158] (Figure 4). Five 3D-structures of GSs and one of BRS are available. Note that all correspond to truncated forms lacking a part of their N- or C-terminal ends. Indeed, it remains difficult to crystallize whole enzymes because of their large size (>150 kDa) and the mobility of their domain V [159]. A calcium ion presumed to be involved in enzyme stability is found at the same position in all structures at the interface between domains A and B [23,124,153,157,158,160].

The catalytic domain A comprises the structural elements constituting the catalytic (β/α)8 barrel, at the top of which is found the catalytic triad (Figure 5). The nucleophile, acid/base and TSS are positioned between strand β4 and helix α4, strand β5 and helix α5, and strand β7 and helix α7, respectively. Two α-helices H1 and H2 localized between strand β7 and helix α7 are conserved and specific to GH70 family. They are connected by a loop named loop A1 showing a dual conformation (open or closed upon sucrose binding) in the branching sucrase GBD-CD2 and DSR-M [23,153] (Figure 6A). Molecular dynamic simulations also suggested interactions between loop A1 and the glucan chain in GTF-SI [161]. From the barrel also emerge different flexible loops: loop A2 between strand β2 and helix α2—and loops B1 and B2 at the N-terminus of domain B between strand β3 and helix α3. All these loops delimiting the catalytic cavity vary a lot in length depending on the enzyme and participate in the enzyme dynamics and promiscuity [162] as well as in glucan elongation [154]. Several complexes of glucansucrases with sucrose, maltose, glucose and glucooligosaccharides bound in the active site have been obtained. 

First, GTF180 and DSR-M structures in complex with sucrose enabled subsites −1 and +1 mapping [23,158] (Figure 6A). The strictly conserved residues of motifs II–IV including the nucleophile (Asp677, DSR-M numbering), the catalytic acid/base (Glu715, DSR-M numbering), the transition state stabilizer (Asp790), the arginine (Arg675) and asparagine (Asn681) of motif II, the tryptophan (Trp717) of motif III, histidine (His789) and glutamine (Gln715) of motif IV and that of motif I (Gln1183) interact with sucrose and a tyrosine (Tyr1127) provides a stacking platform for the glucosyl moiety in subsite −1. Two other residues also contribute to sucrose binding: Gln794 localized on the loop linking strand β1 and helix α1 and Asp1118 from loop A2.

Additionally, complexes of GTF180 and GTF-SI with maltose (a very good acceptor substrate for both proteins) were also obtained enabling the mapping of subsites +1 and +2 [157,160] (Figure 6B). The O6 of maltose non-reducing unit was ideally placed to lead to the formation of panose (the product of maltose acceptor reaction). It is only recently that the first 3D structure of a GS (DSR-M) in complex with its natural product (isomaltotetraose) was obtained, enabling the location of acceptor subsites −1, +1, +2 and +3 [154] and the identification of two tryptophan in loop B2 (Trp624 and Tr717), which provide stacking platforms to the growing glucan chain. Their mutations enabled to transform DSR-M into an isomaltooligosaccharide synthesizing enzyme, thus demonstrating their important role for polymer elongation [154]. In addition, docking of isomaltotriose in the active site of GTF180ΔN suggested the presence of alternative +II′ and +II″ subsites different from the +2 subsite defined by the maltose complex. The different positioning of isomaltotriose compared to maltose highlighted several residues of domain B (Leu940, Leu938, Ala978). Their substitutions resulted in a reduction of the α-1,3/α-1,6 linkage ratio both in the polymer and the oligosaccharides, the latter being produced in higher amounts [158,163]. Interestingly, substitution of the conserved leucine (equivalent to Leu 940) in several GSs (DsrI from *Ln. mesenteroides* NRRL B-1118; DSR-S from *Ln. mesenteroides* NRRL B-1118, GtfI from *S. sobrinus*, GtfG from *Ln. pseudomesenteroides* NRRL B-1297 and ASR from *Ln. citreum* NRRL B-1355) also modified the profile and the yields of the oligosaccharides obtained from sucrose. It also allowed to generate variants producing isomelezitoze (α-D-Glu*p*-(1→6)-β-D-Fru*f*-(2↔1)-α-D-Glu*p*), an osmoprotectant, in yields up to 57% [164]. In ASR, docking experiments also suggested the presence of a +2 subsite and a +2′ subsite highlighting two important amino acids: Trp675 of Motif III and Asp772, respectively. Mutagenesis experiments suggested that a subtle interplay of acceptor binding between these two sites could be at the origin of the α-1,6 and α-1,3 alternance specificity [124]. Finally, a surface binding site relatively distant from the catalytic site but still located in domain A was recently identified in ASR [165]. This site was proven to be functional by biochemical studies and involved in the polymerization reaction. The presence of alternative binding subsites may be encountered in other GS enzymes synthesizing different types of glycosidic linkages in the main chain of the polymer. Resolution of a higher number of 3D structures in complex with natural and longer products should help to confirm these hypotheses.

The domain C consists of a single contiguous segment arranged in eight-stranded β-sheets and is similar to the Greek key motif found in the GH13 family enzymes. It is highly conserved in the family but its function is not yet clear. 

The domain V, also called glucan binding domain, adopts different conformations depending on the crystal structures. Indeed, in DSR-M and ASR it points towards the active center while more extended structures are observed for GTF180, GTFA and GBD-CD2 (Figure 4). SAXS analysis of the whole GTF-180 in solution further revealed different conformations with the domain V either far from or close to the domain A, indicating large scale motions around the domain IV, which is proposed to act as a hinge [158,159]. SAXS analysis of DSR-OK also showed an extended structure of its domain V [166]. Not all GSs and BRSs react in the same way to the deletion of their domain V. A slight decrease in activity is observed for GTFA, DSR-M and GTF-I from *S. downei* MFe 28 [23,94,167] but a significant one for GBD-CD2 BRS, DSR-S and DSR-OK [77,119,168]. This could be related to the more or less strong involvement of this domain in glucan elongation or branching, and to enzyme processivity. A subtle interaction between the domain V and the catalytic domain is likely. It could help to capture polymer chains and maintain them in the vicinity of the active site to promote their extension or branching [23,119,159,161,165]. Repeated sequences are found in this domain and their role has been investigated since the 90s [92,94,166,169,170]. Their partial or total truncations influence the glucan binding ability and also modify the size of the synthesized polymer [77,94,119,123,160,171,172,173,174,175,176,177,178]. In particular, the glucan binding domain was proposed to be a major actor of the processive synthesis of high molar mass dextrans by providing anchoring platform with high affinity for the growing chain [119]. This finding is supported by the recent 3D structures of several GSs in complex with isomaltooligosaccharides or oligoalternans fragments bound in domain V [19,23,165]. These complexes enabled the identification of several sugar binding pockets at the N or C-term part of the domain V, their occurrence being dependent on the enzymes (Figure 7). 

They are well conserved at the structural level and the interactions with sugars is mainly mediated by one or two aromatic residues (Tyr, Trp) and a QxK motif proposed to be a marker of binding ability (Figure 8). The biochemical functionality of these pockets and their importance for polymer elongation have been confirmed by substituting the aromatic residue with an alanine in DSR-M, ASR as well as in DSR-OK [23,165,168]. Interestingly, mutations in some pockets did not impact polymer formation indicating that some of them may not be functional, not accessible or not properly oriented for binding [23]. Cooperativity between the sugar pockets upon glucan binding is likely but has not been demonstrated yet [168]. Further investigations are needed and would greatly benefit from the structural analyses of polymer-associated complexes that are difficult to obtain by crystallography [165]. This would also be important to better make the link between conformational dynamics (large scale motions of domain V) and polymer synthesis that remains enigmatic to date. 

## 4. Enzyme Engineering for Man-Made α-Glucans, Oligosaccharides and Glucoconjugates

The number of GS and BRS sequences available in the databases is increasing incredibly and in parallel their biochemical characterization is also expanding. In this context, what is the benefit of generating artificial catalysts, and if there were good reasons for doing so, what would they be? First, biochemical characterization is expanding at a much slower pace than the acquisition of biochemical data, which remains rather limited compared to the potential offered by these enzymes. Second, extensive profiling of linkage and acceptor specificities is limited only to few representatives. Third, even if GS and BRS promiscuity is broad, glucosylation yields can sometimes be rather low due to competition with natural reactions. Side reaction-products issued from hydrolysis, sucrose isomer formation or natural polymerization can become a real problem especially when considering scaling up and downstream processing. All this can dramatically affect process development. In the same vein, if catalytic efficiency of glucansucrases is often quite suitable for industrial usage [179,180], their stability is often rather low. Indeed, alternansucrase is one of the most stable glucansucrases with a half-life time of 75 h at 30 °C and 6 h at 40 °C [181]. By comparison, the half-life time of GBD-CD2 is of only 10 h at 30 °C and 15 min at 40 °C [112]. This can hamper operability for oligo- or polysaccharide manufacturing. To our knowledge, only one example of rational engineering aiming at improving the thermostability of dextransucrase from *L. mesenteroides* 0326 was recently reported. One double mutant (P473S/P856S) was obtained that showed increased of half-life time at 35 °C (7.4-fold) and catalytic efficiency (2-fold) [182]. However, directed evolution or more rational engineering based on introduction of salt bridge or disulfide bonds have not been attempted yet for increasing GS or BRS stability to temperature or non-aqueous solvents. In addition, hyperthermophilic strains producing naturally highly stable glucansucrases remain to be discovered.

Nevertheless, to adress all these limitations and drawbacks, the last 30 years have seen impressive advances in protein engineering and design to reduce the time required to develop new catalysts with improved performances and desired properties over the parent catalysts. Relevant examples illustrate well the potential of engineering approaches when applied to GSs and BSs.

### 4.1. Engineering the Linkage Specificity to Diversify Glucan and Oligosaccharide Structures

As seen in the preceding section and also excellently described in an exhaustive way in several literature reviews [24,48,179], engineering works on glucansucrases began in the 90s right after the discovery of their sequences [183,184,185,186]. Sequence analysis guided enzyme truncatures [94,187,188,189], generation of hybrid GSs from homologues [173,190] and site-directed mutagenesis. It rapidly appeared that linkage specificity was sensitive to mutations of amino acids located near the catalytic residues and defining the acceptor-binding subsites +1 and +2 in the conserved regions II, III and IV of the domain A. 

In particular, the triplet SEV following the TSS in motif IV and defining the subsite +2 in GTF180ΔN is mostly present in dextransucrases (DSR-S, DSR-M, GTF-S) and mutansucrases (GTF-SI, GTF-I), whereas this sequence is replaced by NNS in reuteransucrases (GTFA, GTFO) and YDA in alternansucrases (*Ln. citreum* B-1355, LBAE-C11, KM20, ABK-1, B-1501 and B-1498 strains) (Figure 3). In the dextransucrase GTFR from *S. oralis*, the motif “RAHDSEV” framing the transition state stabilizer Asp627 was submitted to error prone PCR. A mutant R624G:V430I, isolated from a library of 2000 mutants, showed an enhanced propensity to synthesize insoluble polymers with a majority of α-1,3 linkages either in the linear chain (39%) or at branching points (14%) [191]. In DSRBCB4 from *Ln. mesenteroides* B-1299CB4 and GTF180, the SEV and SNA motifs, respectively, were replaced by the corresponding NNS tripeptide of reuteransucrases [192,193,194]. This led to an increase in α-1,3 linkage specificity for DSRBCB4. Both enzymes acquired α-1,4 linkage specificity. Substitutions of the same tripeptide by different motifs such as NNA, YNA, YDA in GTF180 also resulted in slight modulation of α-1,3/α-1,6 linkage specificity and variation of α-1,4 linkage content in the polymers [193]. The NNS mutant of DsrM from *Weissella cibaria* 10M remarkably produced a reuteran polymer with 52% and 48% of α-1,4 linkages and α-1,6 linkage whereas the parental enzyme synthesized a dextran with only 2% of α-1,4 branching linkages [195]. Targeting the same regions in reuteransucrase GTFA also decreased the α-1,4/α-1,6 ratio changing the reuteransucrase into a dextransucrase [196]. However, switching the YDA motif with the SEV motif in the ASR from *Ln. mesenteroides* NRRLB-1355 resulting in a loss of activity and of polymer synthesizing capacity [119] and single mutations Y768A, Y768F or Y768W did not change significantly the linkage specificity [124]. We can conclude that this tripeptide motif is involved in specificity to a different level depending on the enzyme: the NNS motif seems critical for reuteransucrase specificity whereas the YDA and SEV motif could be more permissive to mutations. Another key residue for the control of linkage specificity is the amino acid located five residues downstream of the TSS (equivalent to T516 in the motif 511DSEVQT of DSR-SΔ4N). A threonine is present at this position in dextransucrase and an aspartic acid in mutansucrases. DsrI is an exception as it shows a threonine at this position but produces a water insoluble glucan. However, mutation of the threonine resulted in an increase of α-1,3 linkages formation for all the mutants except T675Y which had the same profile as the wild-type enzyme [197]. In ASR, there is an aspartate at this position (D772), which was shown to define the +2′ subsite and promote α-1,3 linkage formation.

More recently, the original *Lb. reuteri* TMW 1.106 dextransucrase that produces α-1,4 branched dextran was also mutated downstream the nucleophile (mutants V288P:V291I), the acid/base catalyst (mutants S326N) and the transition state stabilizer (mutant S396N:A398S). These single or double mutations were all combined to generate seven mutants producing novel glucan structures varying in α-1,6/α-1,4 linkage content and arrangement. Their detailed characterization of fragments resistant to enzymatic hydrolysis using methylation, HPAEC-PAD and NMR revealed that they constitute a nice structural diversity to study the relations between the structure and the physico-chemical properties and functionalities of the α-glucans [198].

Overall, if structural data provide good guidance for changing linkage specificity of GSs and BRSs with a limited number of mutations, they also highlight the fact that rational engineering of new GSs is difficult in the absence of complexes with long chain oligosaccharides of very well defined structures. This observation led Irague et al. to propose a combinatorial engineering approach, mixing rational design and directed molecular evolution, to change the linkage specificity of DSR-SΔ4N enzyme. They obtained evolved catalysts producing polymers with different structures [199]. Eight amino acids (Asp306, Phe353, Asn404, Trp440, Ser512, Asp460, His463 and Thr464) were targeted from sequence alignment and structural data (from GTF180 in complex with maltose and sucrose). From a library of 30,000 variants first screened for their ability to produce high molar mass polymers, 300 mutants with altered specificity and synthesizing dextrans with up to 25% of α-1,3 linkages were identified by flow-NMR analysis [200]. Notably, no variants with α-1,2 or α-1,4 bonds emerged from this engineering campaign, indicating that a more profound redesign of the DSR-SΔ4N is likely required. The very stringent linkage specificity of this highly processive enzyme and the reshaping only based on modification of eight residues may also explain this result. In addition, if the NMR screen allowed characterizing the specificity of around 500 variants per day, the throughput remained too low for screening libraries of larger size, hence highlighting the fact that more rapid methods still have to be developed to efficiently apply random approaches. In GTF180, mutations of the amino acid residues Asp1085, Arg1088 and Asn1089 in its α-helix 4 (in which are located Asp460, His463, Thr464 of DSR-SΔ4N) yielded mutants synthesizing higher amounts of α-1,3 branching linkages, some of them also displaying a low α-1,4 linkage specificity [201]. In most mutagenesis studies, increasing the proportion of α-1,3 or α-1,4 linkages relatively to α-1,6 linkages is difficult. This observation could be explained by the fact that Q-1,3 and α-1,4 linkages are shorter and more rigid than α-1,6 linkages.

Finally, another approach based on insertion mutagenesis was recently explored to change the linkage specificity of the dextransucrase from *Ln. mesenteroides* 0326 [202]. This enzyme is more prone to hydrolysis and oligosaccharide synthesis than DSR-SΔ4N and produces dextran with 5% α-1,3 linkages. A library of variants displaying one amino acid insertion either in region II between Ala552 and Val553 (following the nucleophile Asp551, DSRLm0326 numbering), in region IV between Asp662 (the transition state stabilizer) was constructed. Primers were designed to allow insertion of all 20 possible amino acids in the selected sites. Interestingly, different profiles emerged with variants synthesizing polymers with 5–13% α-1,3 linkages, up to 52% α-1,4 linkages and up to 6% α-1,2 linkages. A broader exploration of the sequence space is possible through insertion but also deletion, which make sense to engineer the loops defining the active site. The extension of such approaches would also need setting-up very high throughput screening assays, which remains challenging. A support from computational protein design definitely makes sense in this context [203,204,205].

### 4.2. GS Engineering for Size-Controlled Polysaccharides and/or Enhance Production of Oligosaccharides

The resolution of GS structures in complex with isomaltologosaccharides and/or oligoalternans in the domain V provides a better insight into the interactions between the protein and the carbohydrate at the molecular level. This enabled more accurate structure-guided engineering aiming at controlling the size of the polymer [23,165,168]. In particular, substitution of the stacking aromatic residues with alanine in the sugar pockets of DSR-M and DSR-OK yielded mutants producing polymers of lower molar mass lower compared to the parental enzymes. Remarkably, the double mutant DSR-OK_1-Y1162A-F1228A produced dextrans of 10–13 kg/mol (versus around 10^9^ g/mol for the native enzyme) and used a distributive mode of action instead of a processive one [168]. If mutation or deletion targeting the sugar binding pockets are efficient to reduce the polymer size, the reciprocity (in other words the extension and engineering of the glucan binding domain with the view of increasing the polymer size) is more delicate. This was recently illustrated by swapping the domain V of DSR-OK (producing an HMM polymer) with DSR-M (producing an LMM polymer). With the domain V of DSR-OK, DSR-M-Chimeras were severely altered in their catalytic efficiency even if they produced polymers of slightly higher size. The dynamic and interplay between the catalytic domain and the glucan binding is subtle and difficult to recreate in the state of current knowledge.

Mutations in subsites +1, +2, +3 of GS are another means to enhance oligosaccharide production at the detriment of polymer formation [154,155]. However, as seen in the previous section, these mutations also often cause variation in linkage specificity, together with a decrease in activity and increase in hydrolysis. Surface binding sites distant from the catalytic center and comparable to that found in alternansucrase may also represent good targets for mutagenesis in the future to adjust polymer size. In ASR, alteration of the surface binding site SBS-A1 by mutation of residues Gln700 and Tyr717 increased oligoalternan synthesis by one-third. The presence of distant surface binding sites in domain A is also strongly suspected in the α-1,2 branching sucrases GBD-CD2 [19], they could also be present in other GSs or BRSs and intervene in the polymer chain growth or branching.

### 4.3. Engineering GSs and BRSs for Non-Natural Acceptor Glucosylation

Reactivity of glucansucrases towards sucrose donor substrate is very stringent. Even if these enzymes can use glucoside substituted with a good leaving group at the anomeric position (e.g., glucoside-fluoride or paranitrophenyl glucoside), sucrose is by far their preferred glucosyl donor. In contrast, GSs and BRSs recognize a large number of acceptors which are oligosaccharides or non osidic molecules. In this later case, this natural promiscuity has been exploited since the 60s to modulate the physico-chemical properties (solubility, HLB, redox potential) bioactivity (organoleptic, antioxidant properties) or bioavalability of acceptor molecules. The prerequisite for this reaction to occur is the acceptor recognition, which can be significantly improved through binding pocket engineering. Recently, mutants of GTF180ΔN, focused on positions Leu938, Leu981 and Asn1029 in the subsite +1 and altered in their ability to synthesize high molar mass glucans, were also found improved for the glycosylation of several polyphenol and alcoholic compounds (catechol, resorcinol, hydroquinone, butanol) [206]. Interestingly, a collection of 82 mutants of GTF180ΔN was screened for glucosylation of steviol glycosides. Among them, the mutant Q1140E targeting a residue near the transition state stabilizer, emerged as much more efficient than the parental enzyme to transform rebaudioside A into mono glucosylated products. Remarkably, a quasi-total conversion of 200 mM rebaudioside was obtained by fed-batch reaction yielding 270 g/L of rebaudioside A-glucosides showing a reduced bitterness compared to rebaudioside A [135,207]. The dextransucrase from *Ln. reuteri* TMW 1.106 (specific for α-1,4 branched dextran synthesis) was also assessed for flavonoid glucosylation and showed a poor transferase activity towards these acceptors. Mutation of the residue Leu242 (equivalent to Leu981 in GTF180) generated a mutant more prone to glucosylate quercetin, quercetin-3-O-β-glucoside, rutin, epigallocatechin, gallate, dihydromyricetin, and cyanidin-3-O-β-glucoside [208].

Molecular docking of unnatural acceptors is a good way to suggest mutagenesis targets in the acceptor binding subsites +1, +2 or +3 of GSs and BRSs. By docking quercetin, a poorly soluble flavonoid, four residues (Trp2135, Phe2136, Phe2163 and Leu2166) of the α-1,2 branching sucrase GBD-CD2 from *L. mesenteroides* NRRL B-1299 were systematically found in contact with the flavonoid and chosen as target to construct two libraries comprising a total of around 3000 mutants from which around 200 mutants used sucrose as substrate as shown from a pH-based screening assay on solid medium. Of them, 22 mutants converted more than 70% and up to 99% quercetin from 5 mM quercetin and 292 mM sucrose. The study revealed the potential of the branching sucrase for bulky acceptor glucosylation compared to glucansucrases [40]. Indeed, the glucansucrases from *Ln**. citreum* B-1299 and *Ln. mesenteroides* NRRL B-23192 only converted 23% (of 2 mM quercetin) and 4% (of 9 mM quercetin), respectively [131,209]. Mutants improved for naringenin, luteolin and morin were also identified in this subset. In addition, this small collection was also recently tested for the glucosylation of a lightly protected tetrasaccharide to synthesize precursors of glycosidic haptens mimicking the O-antigen motif found at the cell surface of pathogenic Shigella species. Mutants giving access to distinctive product profiles containing up to four distinguishable glucosylation products (two new products compared to the wild-type, and two products obtained in higher amounts) were identified [137].

An interesting approach was proposed for flavonoid glucosylation to extend the promiscuity of GTF-D from *S. mutans*, the first glucansucrase shown to catalyze the glucosylation of catechin, catechol, 4-methylcatechol, and 3-methoxycatechol [42,43]. Site-saturation mutagenesis targeting the subsite +1 of GTF-D, Tyr418 and Asn469 (equivalent to Tyr430 and Asn481 of GTF180ΔN) was carried out and the mutant library was screened by quenching the fluorescence of coumarin 4-methylumbelliferone (4-MU) through glucosylation. The method enabled to isolate a mutant showing an improved ability for the glucosylation of coumarin 4-methylumbelliferone as well as genistein, daidzein silybin and catechin compared to the parental enzyme [210].

## 5. Outlook

The glucansucrases and branching sucrases of GH70 family are versatile enzymes of great potential to transform sucrose in molecules of added values. A higher number of enzymes will be available in the future thanks to the abundance of genomic data, progress in their bioinformatics annotations and decrease of the price of synthetic genes. However, to efficiently exploit this reservoir, we need to make progress in biochemical characterization of these enzymes. Accelerated multi-criteria evaluation and profiling in terms of transglucosylation/hydrolysis capability, stability, catalytic efficiency, linkage specificity, acceptor specificity, stability, side-product formation would be very useful to make the link between sequence and functions. In addition, sets of data acquired in more standardized ways could help feeding machine learning algorithms with appropriate training sets and test their efficacy to guide and accelerate engineering of tailored enzymes dedicated to specific applications [211]. The development of high-throughput assays is also very appealing to accelerate data acquisition. This would facilitate direct molecular evolution of these enzymes and allow isolation of improved catalysts in libraries of very big size. For example, although progressing, the current assays are not generic enough to rapidly discriminate enzymes showing different acceptor specificities. The development of microfluidic assays would definitely be of interest to lift these barriers. However, most of them rely on fluorescent acceptor molecules with the risk of identifying variants efficient only on the fluorescent derivatives and not on the natural substrate. Mass spectrometry coupled assays would solve this problem, but such developments are still in infancy. Another major limitation is the discrimination of the diverse regio and stereochemistries of the glycosidic bonds that can be formed by GS and BRS variants. Structure-sensitive detection methods such as ion mobility–mass spectrometry for variant screening would be a real breakthrough but this is still highly challenging and requires the acquisition of fragment data from diverse and well-defined oligosaccharide structures to be used as structural reporter [212,213]. Direct imaging methods are progressing and definitely hold promise to circumvent these obstacles [214].

To select a glucansucrase and/or branching sucrase, in a more deterministic way, according to the physico-chemical properties of the α-glucan produced, it is also crucial to pursue investigations on the relations between structures and properties of α-glucans. The current knowledge is limited due to the difficulty linked to the unambiguous structural characterization of the polysaccharide populations (particularly in terms of branching positioning and ramification length). Indirect methods combining NMR, partial acid hydrolysis, acetylation, methylation, SEC-MALLS and enzymatic hydrolysis are used to provide structural information enabling the reconstruction of an average structure. In parallel, the number of solved 3D structures of GH70 family enzymes, which remains low, should increase. These multi-domain enzymes remain difficult to crystallize especially in complex with large size oligosaccharides but with the rapid development of cryo-electron microscopy, we may have access in the near future to more informative high-resolution images of GH70 enzymes bound to longer polysaccharide structures, even during the polymerization reaction. Feeding the databases with new 3D structures of BRS and high molar mass dextran forming dextransucrases will certainly give new insights on structural determinants controlling enzyme activity and specificity. The capture of enzyme structures in different conformations (and at better resolution), will further provide structural information on the mobility of the various loops that delineate the catalytic domain, as well as on the synergy between these motions and those of the domain V. The dynamic of the reaction and of polymer elongation would certainly benefit from such data, which combined with molecular modeling could help to integrate dynamics in protein design. Multiple options exist that are just waiting to be implemented to improve the access to glucansucrases and branching sucrases with desired properties.

Finally, another important criterion to consider for the exploitation of the recombinant enzymes resides in the expression systems used for their production. As mentioned in the first part of this review, natural producers (LAB) often secrete a mixture of GH70 sucrases, which is a disadvantage for targeting well-defined product structures. Recombinant expression is therefore attractive. Most of the studied sucrases are recombinantly expressed in *Escherichia coli* both for their fine biochemical characterization and for in vitro synthesis of gluco-products. Optimization of the expression system used in terms of vectors or strains, and culture conditions (time of induction, growth temperature) can greatly enhance intracellular enzyme production [215]. However, recombinant expression in other microorganisms with GRAS (Generally Recognized As Safe) status is also of interest, notably for developing sucrase-based processes dedicated to food applications. Malten et al. reported in 2005 the first recombinant expression of the dextransucrase DSR-S from *Ln. mesenteroides* NRRL B-512F in *Bacillus megaterium* [216] and production in *Bacillus subtilis* has also been attempted. Recently, Skory and Côté described the first example of efficient glucansucrase secretion by *Lactococcus lactis* LM0230, using a high-copy plasmid with nisin induction. Around 380 mg/L of DsrI enzyme were measured in the culture medium, which was 150-fold higher compared to the production level obtained with the native *Ln. mesenteroides* strain [217]. The progress in strain engineering will certainly facilitate production by the native and GRAS organisms. All these future developments will change the landscape and extend the scope of applications of GSs and BRSs.

## Figures and Tables

**Figure 1 microorganisms-09-01607-f001:**
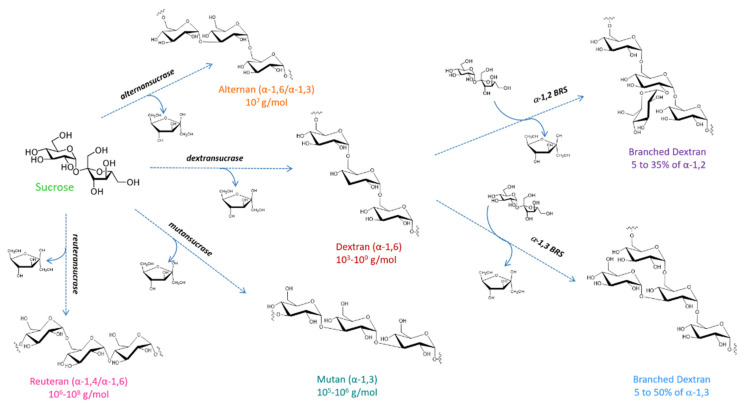
Overview of α-glucan architectures produced by glucansucrases (dextransucrases, mutansucrases, reuteransucrases, alternansucrases) and branching sucrases (α-1,2 BRS and α-1,3 BRS) from sucrose.

**Figure 2 microorganisms-09-01607-f002:**
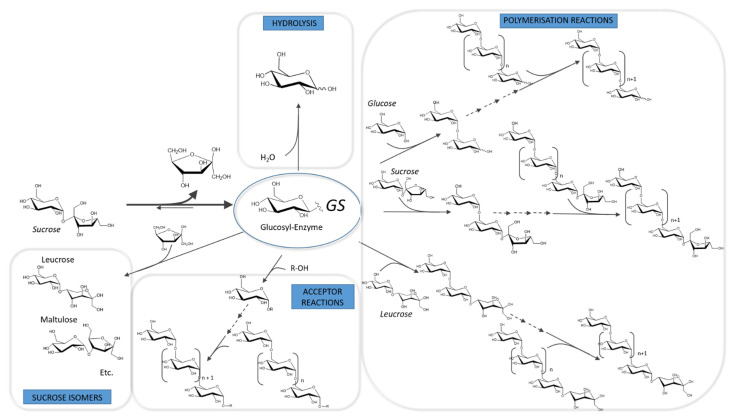
General mechanism of reactions catalyzed by GH70 sucrose-active enzymes: glucansucrases (GSs) catalyze all the reactions described while branching sucrases (BRSs) are dedicated to dextran branching by acceptor reaction.

**Figure 3 microorganisms-09-01607-f003:**
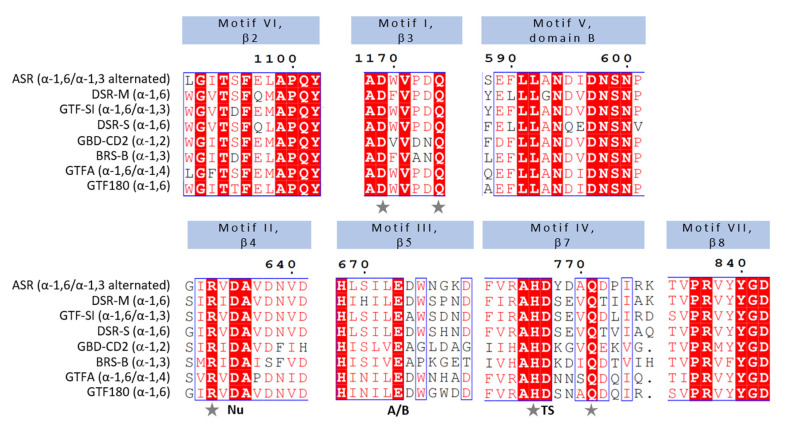
Sequence alignment of highly conserved motifs I-VII in the sequences of sucrose-acting GH70 of available structures. Black stars: catalytic residues. Symbols β2, β3, β4, β5, β7, β8 represent the β strand of the (β/α)_8_ barrel in which each motif is located, the numbering corresponding to that used for the (β/α)_8_ barrel of GH13 family enzymes. Abbreviations Nu, A/B, and TS below the sequence correspond to the nucleophile, general acid/base, and transition state stabilizer, respectively. Grey stars: residues in interaction with the glucosyl moiety in −1 subsite. The alignment was created using ESPript 3.0 [151].

**Figure 4 microorganisms-09-01607-f004:**
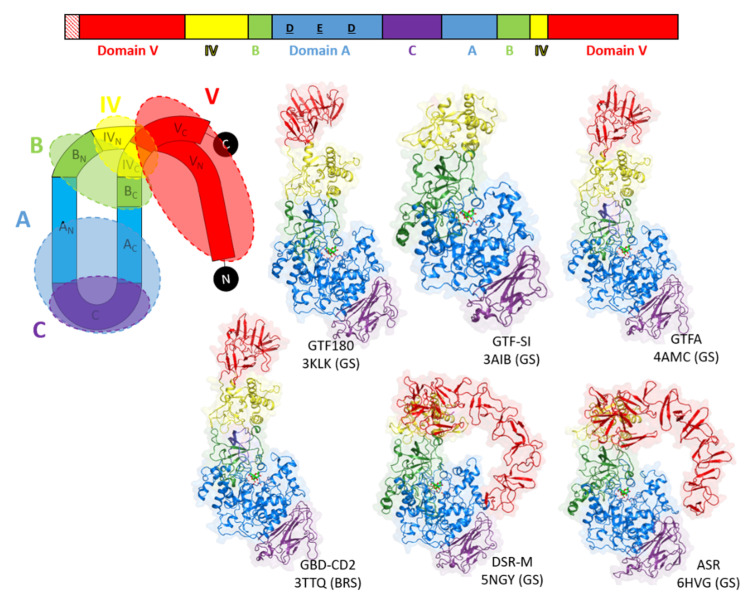
Schematic representation of the U shape fold formed by the various domains of GSs and BRSs (adapted from Vujičić-Žagar et al., 2010 [158] and the six 3D structures available (red: domain V; yellow: domain IV; green: domain B; blue: domain A; purple: domain C). PDB IDs: 3KLK (glucansucrase GTF180-ΔN from *Lb. reuteri* 180, [158]), 3AIB (mutansucrase GTF-SI-catalytic-core from *Streptococcus mutans*, [157]), 4AMC (reuteransucrase GTFA-ΔN from *Lb. reuteri* 1213, [160]), TTQ (branching sucrase ΔN123-GBD-CD2 from *Ln. citreum* NRRL B-1299, [19,153]), 5NGY (dextransucrase DSR-MΔ2 from *Ln. citreum* NRRL B-1299, [23]), 6HTV (alternansucrase, ASRΔ2 from *Ln. citreum* B-1355, [124]).

**Figure 5 microorganisms-09-01607-f005:**
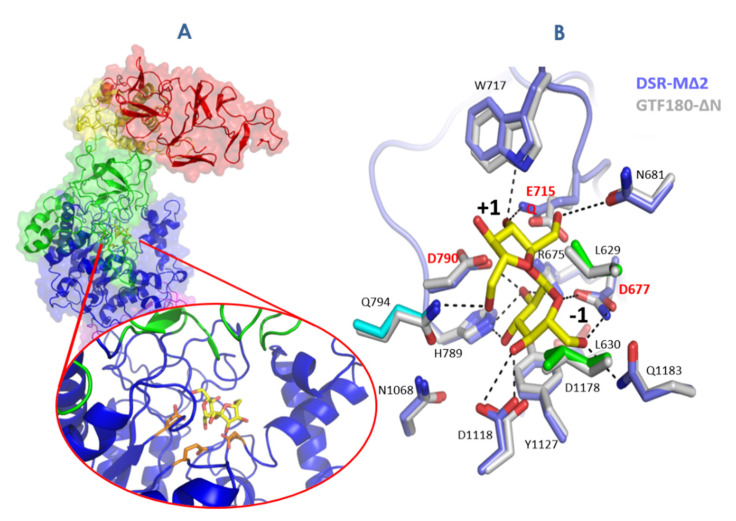
(**A**) 3D structure of an inactive mutant of DSR-M Δ2 in complex with sucrose substrate bound in the active site (PDB: 5O8L, 3.6 Å); (**B**) catalytic site of DSR-MΔ2 and GTF-180-ΔN glucansucrases. Amino acid numbering corresponds to GTF 180-ΔN.

**Figure 6 microorganisms-09-01607-f006:**
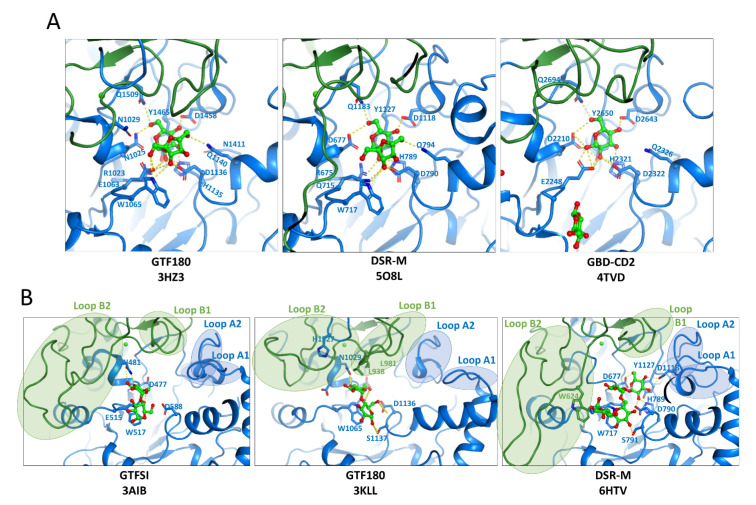
Complexes obtained in (**A**) −1 and +1 subsites of domain A of sucrose-active GH70 enzymes and (**B**) in other subsites (from +1 to +3) in the vicinity of the catalytic site. Loop A1 is defined between the strand β7 and helix α7; loop A2 between the strand β2 and helix α2; loops B1 and B2 at the N-terminus of domain B between strand β3 and helix α3.

**Figure 7 microorganisms-09-01607-f007:**
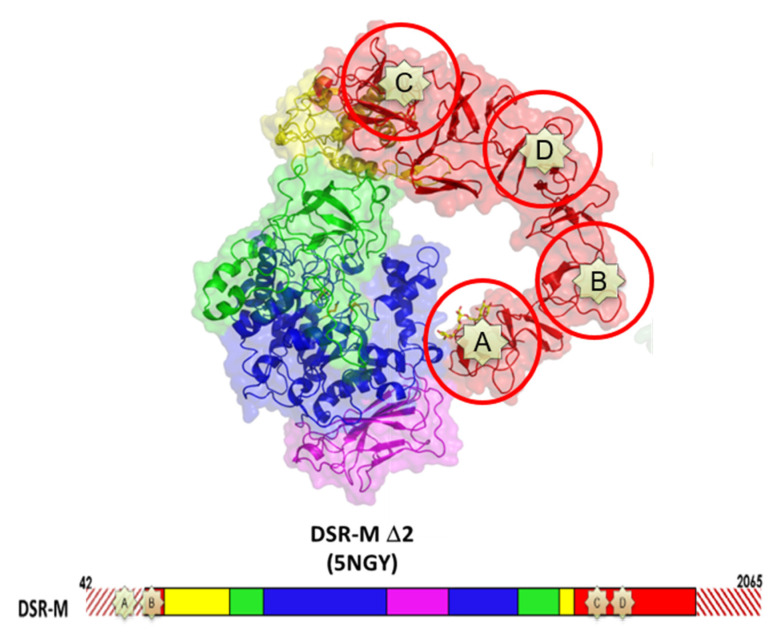
Localization of the sugar binding pockets found in the domain V of DSR-M glucansucrase.

**Figure 8 microorganisms-09-01607-f008:**
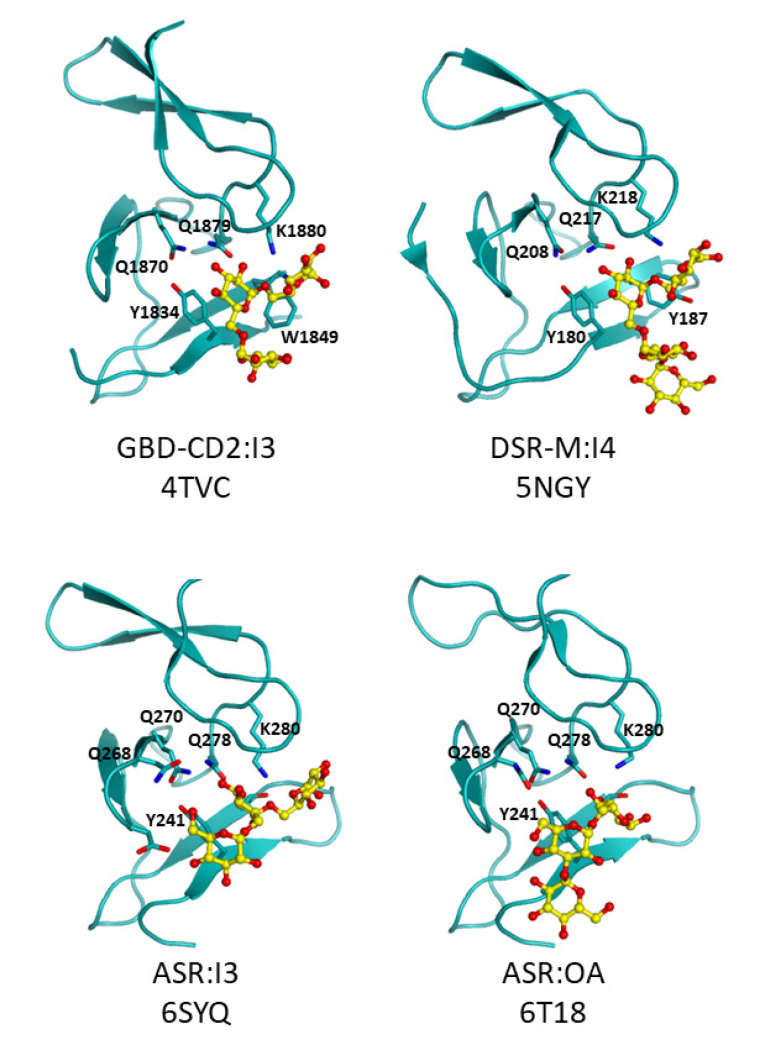
Complexes in sugar binding pockets: GBD-CD2 from *Ln. citreum* B-1299 with isomaltotriose (I3), DSR-M from *Ln. citreum* B-1299 with isomaltotetraose (I4), ASR from *Ln. citreum* B-1355 with isomaltotriose (I3) and an oligoalternan fragment (OA).

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
