# Peer review of "Bacterial α-Glucan and Branching Sucrases from GH70 Family: Discovery, Structure–Function Relationship Studies and Engineering"

_microorganisms, 2021, doi:10.3390/microorganisms9081607_

Round 1

Reviewer 1 Report

The manuscript by Molina and colleagues reviews research on glucansucrase and branching sucrases in GH70. In general, the work is clear and concise. I only have a few minor comments and suggestions.

- The group of prof. Dijkhuizen wrote a review about alpha-glucan synthesis by GH70 enzymes in 2018. Perhaps the authors can consider referring to this earlier review, and mention how their manuscript complements it (e.g. different focus? what changed in the past three years? ...)

- l 141-159: Reading this paragraph made me curious about the phylogenetic relationships in GH70. Are there still a lot of unexplored clades? How (un)related are those recently discovered unusual dextransucrases to the more classical ones? Maybe the authors can discuss this a bit more deeply or add a phylogenetic tree.

- Table 1: I suggest indicating which enzymes have a crystal structure available. Also, it is unclear to me why the amino acid size is mentioned, unless these size differences have structure-functional implications (in which case it is probably better to mention those directly).

- Figure 3: is there a reason why motifs I - IV are in blue and motifs V - VII are in green?

- l418: Please check the symbol before "glucans".

- The authors mention the problem that glucansucrases are often not very stable, but can they discuss possible solutions for readers that may be struggling with this problem? What are the most thermostable glucansucrases/BRS known to date? Has any GH70 enzyme been engineered to increase thermostability?

Author Response

- The group of prof. Dijkhuizen wrote a review about alpha-glucan synthesis by GH70 enzymes in 2018. Perhaps the authors can consider referring to this earlier review, and mention how their manuscript complements it (e.g. different focus? what changed in the past three years? ...)

Answer: As requested the reference to Gangoiti et al, Biotechnology advances, 2018 (https://doi.org/10.1016/j.biotechadv.2017.11.001 ) has been introduced together with that of Meng et al also from the group of Lubbert Dijkuizen on page 3 line 96. In addition, we attempted to make it clearer that our review focuses on the most recent advances in the discovery, mechanistic study and engineering of GH70 family glucan and branching sucrases only. 

“The amazing versatility of GSs and BRSs holds multiple assets for the development of new biosourced molecules targeting an even wider range of innovative applications. Nowadays, this potential is clearly amplified by the discovery of an increasing number of new enzymes, by the progress made in the understanding of their molecular mechanism and by engineering approaches that can be deployed to expand their synthetic capabilities. This review places the focus on these various aspects and intends to cover more specifically the most recent and exciting developments.”

Clarifications are highlighted in green in the revised version.

- l 141-159: Reading this paragraph made me curious about the phylogenetic relationships in GH70. Are there still a lot of unexplored clades? How (un)related are those recently discovered unusual dextransucrases to the more classical ones? Maybe the authors can discuss this a bit more deeply or add a phylogenetic tree.

Answer : If we refer to the sequences available in the CAZy database (around 800) compared to the number of enzymes biochemically characterized, there is a lot of enzymes to be discovered that belong to different clades. However, we felt that introducing a phylogenetic tree did not bring additional information compared to that recently published in Gangoiti et al, Biotech. Adavnces 2018.

- Table 1: I suggest indicating which enzymes have a crystal structure available. Also, it is unclear to me why the amino acid size is mentioned, unless these size differences have structure-functional implications (in which case it is probably better to mention those directly).

Answer: Fixed. An asterix indicates which enzymes have a crystal structure available.

- Figure 3: is there a reason why motifs I - IV are in blue and motifs V - VII are in green?

Answer: There was no specific reason and for a sake of clarity we have modified this figure. For each motif, we have added the b strand in which they are located. The ß strand  number corresponds to the numbering used for the (ß/α)8 barrel of the GH 13 family enzymes.

- l418: Please check the symbol before "glucans".

Answer : Fixed.

- The authors mention the problem that glucansucrases are often not very stable, but can they discuss possible solutions for readers that may be struggling with this problem? What are the most thermostable glucansucrases/BRS known to date? Has any GH70 enzyme been engineered to increase thermostability?

Answer : As requested we have given more details on the stability of GSs and BRs and  precised that there is only one rational and no random engineering approaches described to date to improve their stability.

Please see Page 13 line 454 and below the modified text:

“In the same vein, if catalytic efficiency of glucansucrases is often quite suitable for indus-trial usage [180,181], their stability is often rather low. Indeed, alternansucrase is one of the most stable glucansucrases with a half-life time of 75 hours at 30°C and 6 hours at 40°C [182]. By comparison, the half-life time of GBD-CD2 is of only 10 hours at 30°C and 15 minutes at 40°C [112]. This can hamper operability for oligo- or polysaccharide manu-facturing. To our knowledge, only one example of rational engineering aiming at improv-ing the thermostability of dextransucrase from L. mesenteroides 0326 was recently reported. One double mutant (P473S/P856S) was obtained that showed increased of half-life time at 35°C (7.4-fold) and catalytic efficiency (2-fold) [183]. However, directed evolution or more rational engineering based on introduction of salt bridge or disulfide bonds have not been attempted yet for increasing GS or BRS stability to temperature or non-aqueous solvents. In addition, hyperthermophilic strains producing naturally highly stable glucansucrases remain to be discovered”

Reviewer 2 Report

Dear Editor,

In the review entitled “Bacterial α-glucan and branching sucrases from GH70 family: Discovery, Structure-function relationship studies and Engineering” Molina and co-authors describe the glucansucrases and branching sucrases belonging to glycoside hydrolase 70 family from a structural and functional point of view. Moreover, the Authors describe the advances in enzyme engineering to develop new GH70s with desired properties.

Overall, the Authors carried out a good bibliography research and cited several recent works. However, this review contains a lot of information and examples that make the text poorly focused and cryptic for non-expert readers. To better understand the section 3 and 4 is required a more accurate description of the GH70 architecture and 3D structure, including a figure indicating the remarkable elements of the enzyme such as active site, sugar binding site, motifs and site -1, +1 etc. An overview of the GH-H clan is required to understand the differences and analogies with the GH13.

This review should be reorganized making it suitable for a non-expert reader. This could also be achieved by using a more schematic structure and a focused writing style.

Analytical comments:

Pag. 2 Line 74. The difference between glucansucrases and branching sucrases is not clear. Please add more information and indicate the differences between these two classes of enzymes, which belong to the same GH family.

Pag. 2 Line 76. Glycoside hydrolases catalyze hydrolysis reactions, however glucansucrases are mainly active in synthesis reactions. Please clarify this point and add more information on the physiological role of GH70.

Pag. 3 Line 102. The section 2 of this review is rather obscure for a non-expert reader. It is not clear the differences between native and recombinant GS and BRS. Moreover, this section contains a lot of information that deserves to be better organized. I suggest re-writing this paragraph in a more schematic and focused way, to make the central message clearer.

Pag. 4 Line 141. The overview of GH70 family should be moved at the begin of section 2.

Pag. 4 Line 171. The authors have not yet described the GH70 architecture. Therefore, it is not clear whether the presence of two catalytic domains is a peculiarity or not. This paragraph should be moved to a later section.

Pag. 8 Line 247. I wonder why the GH70 architecture is described in the paragraph entitled “Catalytic mechanism & products”.

Pag.8 Line 258. The Authors introduced here the conserved motif; however, it is unclear the localization of this motif along the sequence. In addition, these motifs are not represented in the scheme in Figure 4. Overall, the description of architecture of GH70 is important to understand the structural features of this enzyme family and should be introduce earlier (i.e., in the introduction of section 3).

Pag. 9 Line 288. In this paragraph, the Authors should add the architecture of DSR-E enzyme from Ln. citreum, which is characterized by two catalytic domains.

Pag. 13 Section 4. This section is reach of examples indicating how protein engineering can improve the properties of GH70. Unfortunately, the result is a long list of enzymes and mutated amino acid residues. I suggest reorganizing this paragraph by giving a general overview of the effect of these mutations. Is it possible to rationalize the effect of some types of mutations? Furthermore, these mutations could be located in the 3D structure of the enzymes to facilitate reading.

Minor comments:

In many cases the species of organisms are not in italics, please check and correct.

Pag. 2 Line 46. Food should be food.

Pag.2 Line 77. Please add the abbreviation of Glycoside hydrolases.

Pag. 10 Line 367. Please add the Figure number.

Author Response

To better understand the section 3 and 4 is required a more accurate description of the GH70 architecture and 3D structure, including a figure indicating the remarkable elements of the enzyme such as active site, sugar binding site, motifs and site -1, +1 etc. An overview of the GH-H clan is required to understand the differences and analogies with the GH13.

Answer :

We hope the reviewer will understand that the aim of the paper was not to give an overview of the GH H clan because this has been extensively described in many other excellent reviews cited in our article.

Regarding  the link with the GH13 family, also extensively described in Meng et al 2016 and Gangoiti et al 2018 and for a sake of clarity, we have modified the figure 3 in which now we have reported the b-strand of the (b/a)8 barrel using the numbering used in the GH13 family. This gives, we hope, a clearer vision of the links between GH13 and GH70 family. AS the topic of our paper is not the evolution between GH13 and GH70 family, we did not comment further.  

This review should be reorganized making it suitable for a non-expert reader. This could also be achieved by using a more schematic structure and a focused writing style.

Answer : As requested we have added two other figures: one showing the location of the -1 and +1 subsites (Figure 5) and the other showing the sugar binding pockets in the domain V (figure 7).

Analytical comments:

Pag. 2 Line 74. The difference between glucansucrases and branching sucrases is not clear. Please add more information and indicate the differences between these two classes of enzymes, which belong to the same GH family.

Answer:

Page 2 Line 75: The following sentences have been added to clarify the function of BRSs compared with that of GSs.

Text added in the revised version

“Indeed, BRSs are specific for linear a-1,6 linked dextran branching to produce highly branched polymers (Figure 1) [19–21], and in the absence of dextran acceptor, BRSs almost exclusively catalyse hydrolysis.”

Pag. 2 Line 76. Glycoside hydrolases catalyze hydrolysis reactions, however glucansucrases are mainly active in synthesis reactions. Please clarify this point and add more information on the physiological role of GH70.  

Answer:

Indeed, GH70 enzymes are often very efficient transglycosylases, this a fact. They have probably evolved in this way from GH13 enzymes to give an environmental advantage to the LAB producing these enzymes, the physiological of these polymers is described in page 1 line 29 with references. Besides, the physiological role of GSs and BRSs remains putative and has not been investigated further. Consequently, we did not comment further.

Text related to the physiological role of alpha-glucans and of glucansucrases (page 1 line 29):

“In this field, a-glucans produced by Lactic Acid Bacteria (LAB) from different genera (mainly Streptococcus, Leuconostoc, Lactobacillus, Weissela) when grown on sucrose are particularly appealing targets [7–9]. These polymers participate in biofilm formation and can protect the bacteria against environmental stresses (e.g. dessication, biocides, antibiotics, phagocyte attack). They can also mediate the adhesion to surfaces [10]. In particular, a-glucans formed by the Streptococcus genus were well-studied due to their contribution to the dental plaque formation, the tooth surface colonization and the development of dental caries [11].”

Pag. 3 Line 102. The section 2 of this review is rather obscure for a non-expert reader. It is not clear the differences between native and recombinant GS and BRS. Moreover, this section contains a lot of information that deserves to be better organized. I suggest re-writing this paragraph in a more schematic and focused way, to make the central message clearer.

Answer:

Among the characterized enzymes, we have decided to distinguish the native enzymes produced by LAB from the recombinant one produced by E. coli. This can be important for end users that do not want to develop processes with recombinant enzymes. This has been clarified in the title of each paragraph of section 2. The message here was just to give the reader an overview of all the characterized GSs and BRS. Without any other guidelines, we hope the reviewer will understand that we did not rewrite the entire section.

Pag. 4 Line 141. The overview of GH70 family should be moved at the begin of section 2.

This section was placed here to emphasize on the fact that production of recombinant enzymes was essential to enable the isolation of single enzyme and to introduce the second part dedicated to recombinant enzymes.

Pag. 4 Line 171. The authors have not yet described the GH70 architecture. Therefore, it is not clear whether the presence of two catalytic domains is a peculiarity or not. This paragraph should be moved to a later section.

Answer:

We have modified the text and specified that only two enzymes showing two catalytic domains have been described to date. We did not move the paragrph because one of these enzymes has been very recently discovered.

Modified text:

“Finally, two enzymes containing two catalytic domains both featuring the characteristics found in GH70 family have been described. In these enzymes, one domain shows GS activity and the other one BRS activity.

Pag. 8 Line 247. I wonder why the GH70 architecture is described in the paragraph entitled “Catalytic mechanism & products”.

Answer : Indeed, the GH70 architecture should be placed at the beginning of section 3.2 entitled Mechanistic insights from structure-based information

Pag.8 Line 258. The Authors introduced here the conserved motif; however, it is unclear the localization of this motif along the sequence. In addition, these motifs are not represented in the scheme in Figure 4. Overall, the description of architecture of GH70 is important to understand the structural features of this enzyme family and should be introduce earlier (i.e., in the introduction of section 3).

Answer:

To improve the description of the architecture of the GH 70 sucrasesFigure 3 (with the motifs) has been improved to better visualize the circular permutation occurring between GH13 and GH70 enzymes. We have precised the position of the motif with regard to the b-strands of  the (b/a)8 catalytic barrel (Domain A). In addition, Figure 5 and Figure 7 have been added to better describe the structure of the GH70 sucrases, and better locate the -1 and + 1 subsites (Figure 5) and the sugar binding pockets in the domain V of these enzymes (Figure 7).  

Pag. 9 Line 288. In this paragraph, the Authors should add the architecture of DSR-E enzyme from Ln. citreum, which is characterized by two catalytic domains.

We have published many papers on DSR-E and did not feel it was central for this paper. In addition, we preferred to add Figure 5 and 7 than putting a scheme of DSR-E that did bring much information.

Pag. 13 Section 4. This section is reach of examples indicating how protein engineering can improve the properties of GH70. Unfortunately, the result is a long list of enzymes and mutated amino acid residues. I suggest reorganizing this paragraph by giving a general overview of the effect of these mutations. Is it possible to rationalize the effect of some types of mutations? Furthermore, these mutations could be located in the 3D structure of the enzymes to facilitate reading.

Answer : It is very difficult to rationalize the effect of these mutations more than what was attempted in the paper. In addition, it is not possible to locate all of them on the structure because the figure would not be readable. We hope the reviewer will understand our position.

Minor comments:

In many cases the species of organisms are not in italics, please check and correct.

Answer : Fixed

Pag. 2 Line 46. Food should be food.

Answer : Fixed

Pag.2 Line 77. Please add the abbreviation of Glycoside hydrolases.

Answer : fixed

Pag. 10 Line 367. Please add the Figure number.

Answer : fixed

Round 2

Reviewer 2 Report

Dear Editor,

the Authors responded properly to all comments.